# Leadership and the Promotion of Health and Productivity in a Changing Environment: A Multiple Focus Groups Study

Julio Miño-Terrancle [1], José M. León-Rubio [2], José M. León-Pérez [2,*] and David Cobos-Sanchiz [3]

1   Department of Work and Social Security Law, Universidad Pablo Olavide, 41013 Sevilla, Spain; jminter@upo.es
2   Carmides Research Lab, Department of Social Psychology, Universidad de Sevilla, 41018 Sevilla, Spain; jmleon@us.es
3   Department of Education and Social Psychology, Universidad Pablo Olavide, 41013 Sevilla, Spain; dcobos@upo.es
*   Correspondence: leonperez@us.es; Tel.: +34-954-557-707

**Abstract:** Leaders that focus on preventing risks and promoting safe and healthy behaviors are essential to reducing workplace accidents and illnesses, particularly in a changing environment where technology and the complex interconnection of systems create emerging risks with unpredictable consequences for employee wellbeing and organizational productivity. In that sense, this multiple focus group study with 32 experts in occupational safety and health (OSH) aims at providing valuable insight into the most effective strategies for promoting health and productivity in a changing context. Results indicate that a safety and prevention culture is crucial for successful risk prevention and management, with commitment required from both top management and workers. Moreover, transformational leadership is identified as a key to achieving a safety and prevention culture. In addition, training is considered a pivotal mechanism to introduce appropriate safety practices into daily work routines. This requires an interdisciplinary, integrated, and collaborative perspective. Finally, integrating risk prevention into higher education prepares professionals to face current labor market challenges. These results can guide decision making for both training OSH professionals and introducing effective OSH practices in organizations.

**Keywords:** occupational risk prevention; transformational leadership; safety culture; decision making; higher education

## 1. Introduction

Accidents and diseases at work have devastating effects on workers, businesses, and entire communities and economies. While many improvements have been made in recent years, the prevention of accidents and illnesses related to work remains a concern on a global scale. The International Labour Organization expressed in June 2022, during the International Labour Conference, that organizations should promote a safe and healthy working environment, as well as prevent or reduce work-related risks by anticipating measures at each stage of its work processes.

Therefore, in the last years, several authors have pointed out that occupational risk prevention (ORP) needs to be integrated into business management at all levels and in all departments, so that it becomes a habitual and systematic practice [1–4]. Indeed, integrating ORP involves not only compliance with legal regulations and standards regarding occupational safety and health (OSH), but also a proactive and committed attitude of the company and its workers to identify, evaluate, and manage ORP [1–4].

As various public bodies and researchers insist, developing a safety and risk management culture is critical for reducing accidents and promoting safe and healthy work practices [5]. For example, the National Institute for Occupational Safety and Health (NIOSH), a research agency part of the U.S. Department of Health and Human Services,

have developed the Total Worker Health (TWH) program, which combines practices that integrate protection from work-related safety and health hazards with the promotion of injury and illness prevention efforts to advance worker wellbeing. In a similar vein, the International Organization for Standardization (ISO), a nongovernmental international organization that brings together experts to develop international management standards, also has a holistic approach in its ISO45001:2018 standard that specifies requirements for an occupational health and safety management system (OHSMS).

Furthermore, several studies have demonstrated that these integrative approaches have a positive impact not only on reducing work-related injuries and illnesses, but also on employee performance and organizational productivity through improved psychological wellbeing [6–8]. For example, Cooklyn and colleagues [9] conducted a systematic review of 31 studies and concluded that such studies classified as "high" quality showed promising effects on several indicators of the psychosocial work environment, including improved job quality, reduced occupational stress, reduced symptoms of depression, and improved psychological resources. Thus, as the NIOSH motto says, occupational health and safety management systems can help in promoting productive workplaces.

Unfortunately, despite OSH being established as a professional discipline and a universal labor right in the European Union (EU) through a Framework Directive (Directive 89/391/EEC-OSH) [5], there are too many risk factors present in organizations because of the lack of a solid preventive culture, which causes occupational accidents to remain at unacceptable levels [6–8].

One of the factors that may explain the lack of a solid preventive culture is the fact that OSH professionals do not have the necessary expert knowledge. For example, a recent study in the Spanish construction sector showed that less than 50% of trainers in OSH have the required expertise in the field [9]. Therefore, it is essential that trainers are highly trained in OSH to ensure that their training is effective and contributes to preventing occupational accidents [10].

We are aware that occupational accidents are due to a series of complex and varied causes, including sociocultural, economic, and political factors both external and internal to the organization [11]. However, all are reducible, to some extent, to values, attitudes, and behaviors shared by the actors involved (i.e., preventive culture). In that sense, more research is needed on the variables that can modify or positively influence attitudes and behaviors related to OSH [12], particularly in contexts where public policies, actions, and incentives to promote a preventive culture are insufficient due to malpractice and corruption [13].

To address the problem of the lack of attitude towards ORP, it is necessary to prioritize adequate training that promotes the required cultural change in organizations, in which senior management plays a leading role by exemplifying and shaping employees' behavior [14,15]. This cultural change must be based on values and beliefs shared by all members of the organization, which shape intimately assumed attitudes, thoughts, and feelings and determine the organization's commitment, style, and skill to fulfill its mission, shaping what we understand by a prevention culture [16,17]. Furthermore, in line with Article 6.3a of the EU Framework Directive, it is necessary to address the study of the requirements for preventive integration, focusing on the relationship between knowledge management and prevention [18–20].

Therefore, our research seeks to answer the following question: How can business advisors and leaders make adequate technical and moral decisions to establish flexible processes and procedures that, considering changing physical, social, and economic conditions, effectively and simultaneously address the premises of health and productivity?

Exploring how leaders can integrate preventive management into their decision making is an important step in addressing the problem of the lack of a preventive culture. Moreover, previous studies have explored how preventive management and leadership are related and how this positively affects OSH and, thus, worker performance and productivity [21,22]. However, unlike other studies, the objective of this qualitative study through

multiple focus groups is to explore how managers and OSH experts consider that OSH services can be integrated into the company's management system.

In other words, this research aims to gain insight into the perspectives and attitudes of decision makers regarding the incorporation of safety and health measures into the organizational structure of their companies. By understanding how these stakeholders view the integration of OSH services, we can identify potential barriers and opportunities for improving the effectiveness of OSH practices in the workplace and, in turn, improving organizations' health and productivity.

*Leadership and the Integration of the Preventive System in Companies*

To achieve an effective integration of the preventive system in companies, it is necessary to overcome the idea that "the human condition is unalterable, but the conditions in which people work can be changed" [23]. This is a limiting and not rigorous idea, as it ignores human capacity to adapt and evolve, as well as the influence of a wide variety of factors on working conditions, many of which are modifiable in the very long term. Furthermore, it ignores the interdependence of both conditions, as evidenced by demand–control and effort–reward theoretical models [24–27].

It is essential to have a broader framework that addresses the different factors that affect workers' safety and health and allows the integration of medium-range theses into its theoretical structure that facilitate more precise explanations of the relationship between specific variables in specific contexts, such as OSH.

We consider the application of organizational learning theory [28], knowledge management models [20], the idea of personal and organizational resilience [29–31], and organizational culture theory [32] to be useful for our objective (see below) because they highlight the importance of continuous learning, knowledge management, and organizational resilience in creating a strong and effective organizational culture that enables organizations to adapt and thrive in a changing environment.

Empirical studies have found evidence of the effectiveness of these theories in improving organizational performance and adaptation. For example, organizational learning can improve the ability of organizations to adapt to changes in the environment, knowledge management can contribute to improving organizational efficiency and effectiveness through the creation, storage, and effective use of knowledge, and resilience has the potential to improve the ability of organizations to withstand and recover from critical situations. Finally, shared values, beliefs, and norms in an organization can have a beneficial effect on its functioning and outcomes.

In this organizational context, leaders have a crucial role in managing occupational risk and integrating it with corporate or organizational management. It is important for leaders to develop adaptability that allows employees to work in harmony and without distress situations that can weaken their performance [3,22]. Many examples can be drawn from the relevant literature on occupational stress or harassment and violence in the workplace [33,34], but we prefer to devise one that highlights the role of leadership as a resilient strategy, such as the transition to teleworking during the pandemic. In this situation, the leader can communicate the need for change and provide resources and tools for effective work from home. They can also establish clear protocols for online communication and collaboration to help employees stay connected and productive. In addition, the leader could demonstrate flexibility and understanding by acknowledging that employees may be facing personal challenges during the pandemic, such as caring for children or sick family members, and offering support and flexibility so they can balance their work and personal responsibilities. Overall, a leader who demonstrates adaptability during the transition to teleworking can help create a more stable and productive work environment for employees, which can in turn improve company performance.

The leader's ability to adapt to various circumstances and conditions is a strong argument for conceptualizing leadership as a resilient strategy, which is fundamental in situations of crisis and uncertainty, as highlighted by the recent COVID-19 pandemic [35–38].

To achieve this, a culture of collaboration, communication, and support in the company must be fostered, where learning is promoted at all levels, and self-criticism and the use of the results that occur because of interactions between different positions and levels of the organization are modulated.

To achieve this, it is necessary that leaders promote a healthy and productive work environment that allows employees to adapt to the variability of work and maximize their capacity for collaboration, communication, and support to achieve the organization's objectives. In that sense, proper management of preventive leadership is considered key to addressing the problem of the lack of a culture of prevention in organizations, often considered the root cause of accidents, occupational diseases, and pathologies, making its promotion essential [39–43]. However, companies are dynamic and complex constructions, and preventive knowledge has a transdisciplinary nature [44,45].

This entails facing a number of challenges, as yet uncalculated, among which we highlight those that have implications for decision making by leaders of any type of organization: the effectiveness of prevention in a context characterized by the evolution of technology, globalization, the interconnection of complex systems, etc., which point to potential benefits but also highlight a series of emerging risks, whose consequences are unpredictable and often characterized by high uncertainty and a lack of reliable data. This situation requires strengthening capacities such as resilience and new modes of prevention management, both of which require a strong preventive culture as a basic pillar of education and knowledge management, both in its academic aspect and in the field of business practice [12].

Therefore, this study aims to provide recommendations and guidance on how business advisors and leaders can make adequate technical and moral decisions to establish flexible processes and procedures that consider changing physical, social, and economic conditions, and effectively and simultaneously address the premises of health and productivity. In summary, this study seeks to identify best practices and strategies to promote an optimal balance between health and productivity in a constantly changing business environment.

Furthermore, the study seeks to promote collaboration and knowledge exchange among managers, researchers, and professionals to develop innovative and effective solutions that address current business challenges. Likewise, the study can help raise awareness among business leaders about the importance of considering both technical and moral considerations when making decisions that affect the health and productivity of their employees and the company.

## 2. Materials and Methods

### 2.1. Study Design and Participants

We used focus groups to explore the perception and experience of OSH researchers and practitioners, business managers, and political decision makers regarding how leaders could establish flexible processes and procedures that effectively address OSH requirements, considering changing socioeconomic and technological conditions, and integrate OSH into management systems.

The study was approved by the Ethics Committee of the Scientific Association of Experts in Safety and Health at Work in Andalusia (ACESSLA) (protocol code SEJ458, approved on 25 March 2020), and all procedures were conducted in accordance with the Declaration of Helsinki of 1964 and its later amendments. Written informed consent was obtained from all participants before conducting the focus groups. The participants were selected through purposive sampling. We selected 32 professionals among the members and collaborators of the Scientific Association ACESSLA, who met the following inclusion criteria: (1) have proven experience in management, research, or technical aspects of OHS, and (2) have formal accreditation of the knowledge, skills, and abilities required to perform activities related to OSH.

We contacted each of them by phone and email to request their participation, gather their availability to participate in the study, and obtain their informed consent. All accepted

and were distributed into focus groups (see Table 1). Thus, we conducted four focus groups (i.e., multiple focus groups that are optimal for saturation responses, see [46]) that discussed the following four topics that have been identified in previous studies as crucial for integrating OSH into management systems [1–12]: (1) preventive effectiveness in the context of the future of work and emerging risks. The recommendations and guidance discussed in this section can help address the challenges faced in managing OSH in a constantly evolving world; (2) tools for strengthening resilience and improving a preventive culture. The discussion on this topic can help leaders develop valuable skills and effective strategies to face the challenges of leadership in an increasingly complex and unpredictable world; (3) reforms and implementations necessary for a comprehensive undergraduate, postgraduate, and continuing education in occupational health. Comprehensive education in occupational health is a key element for business leaders to make appropriate technical and moral decisions in managing health and productivity at work; and (4) training and information in the company for an integrated training of the human factor. Training the human factor is essential to ensure the effective implementation of any OHS process or procedure in an organization. Company training and information are crucial to ensure that workers are prepared to effectively comply with these processes and procedures.

**Table 1.** Characteristics of the participants according to panel of adscription.

| Characteristic | | Panel A | Panel B | Panel C | Panel D | Total |
|---|---|---|---|---|---|---|
| N participants (% relative) | | 8 (23.5) | 9 (26.5) | 9 (26.5) | 8 (23.5) | 34(100) |
| Age Range | 25–34 | 0 | 1 (11.1) | 0 | 1 (12.5) | 2 (5.9) |
| | 35–44 | 5 (62.5) | 0 | 0 | 2 (25) | 7 (20.6) |
| | 45–54 | 0 | 6 (66.7) | 5 (55.6) | 0 | 11 (32.3) |
| | 55–64 | 3 (37.5) | 2 (22.2) | 4 (44.4) | 5 (62.5) | 14 (41.2) |
| Profession | Directors and Managers | 2 (25) | 5 (55.6) | 1 (11.1) | 6 (75) | 14 (41.2) |
| | STEM Technicians [1] | 4 (50) | 4 (44.4) | 0 | 2 (25) | 10 (29.4) |
| | University professors | 2 (25) | 0 | 8 (88.9) | 0 | 10 (29.4) |
| Economic Sector | Secondary | 3 (37.5) | 3 (33.3) | 0 | 3 (37.5) | 9 (26.5) |
| | Tertiary | 5 (62.5) | 6 (66.7) | 9 (100) | 5 (62.5) | 25 (73.5) |
| Work Area | Corporate | 3 (37.5) | 8 (88.9) | 0 | 7 (87.5) | 18 (52.9) |
| | Public Administration | 5 (62.5) | 1 (11.1) | 0 | 1 (12.5) | 7 (20.6) |
| | Public University | 0 | 0 | 9 (100) | 0 | 9 (26.5) |
| Experience in OHS | Policy and Management | 2 (25) | 4 (44.4) | 1 (11.1) | 6 (75) | 13 (38.2) |
| | Job Design and Risk Prevention | 1 (12.5) | 3 (33.3) | 0 | 0 | 4 (11.8) |
| | Promoting a Culture of Prevention | 2 (25) | 0 | 8 (88.) | 0 | 10 (29.4) |
| | Supervision, Evaluation, and Training | 3 (37.5) | 2 (22.2) | 0 | 2 (25) | 7 (20.6) |

[1] "STEM Technicians" is an acronym that refers to technicians who work in the fields of Science, Technology, Engineering, and Mathematics (STEM).

Each focus group was moderated by a researcher previously trained in the technique of focus groups. For each focus group, a discussion guide was prepared with open-ended questions that allowed exploring the participants' perceptions on the topic they were supposed to discuss, considering the main objective of the study. These guides were prepared after a review of the relevant legislation and an analysis of the influential literature related to the issue raised.

The following phases were established for the development of the focus groups: (1) presentation of the study topic and objectives of the focus group; (2) group discussion about it and the research questions; (3) discussion about the participants' personal opinions and experiences related to it; and (4) conclusions and closure of the focus group.

All group sessions were held at the University facilities, and each lasted approximately three hours, with two 20 min breaks in between. Three weeks after the closure of the focus groups, we sent the participants a summary of the debate and conclusions and asked for corrections to increase reliability [47]. The participants provided small suggestions that we incorporated before the analyses.

*2.2. Data Analysis*

All sessions were recorded on video and subsequently transcribed into text using the Sonix tool. The text edition was reviewed to improve the accuracy of the transcriptions, since the participants' pronunciation was not always clear enough during some moments of the recording. A total of 164 pages were obtained, divided as follows: 37 from panel A, 45 from panel B, 47 from panel C, and 35 from panel D, including summaries from the coordinators of each panel. Additionally, in the transcriptions, any mention of the individual speaking, as well as references to their peers by name, were replaced with a unique participant identifier (ID). This ID includes a letter corresponding to the panel they were part of and a number indicating their individual contribution. This measure ensures the confidentiality of the participants while still allowing the origin of each contribution to be tracked. Subsequently, for publication purposes, the text was translated into English using the mentioned tool. To ensure accuracy and fidelity to the original content, a process of revision and editing took place after the initial translation, where a professional reviewed the text to ensure coherence with the original intent and the absence of grammar or spelling errors. These combined methods ensured a precise and faithful translation of the original content.

To analyze the transcripts of the focus groups, we used computer-assisted inductive content analysis, employing QDA Miner Lite software v6, Provalis Research, Montreal, Canada. Like other focus groups studies in the field [48–53], two team members identified and labeled relevant text fragments to address the research question. The obtained codes were grouped into broader categories based on their similarities and differences and were refined and adjusted iteratively as they were compared and discussed with the other two team members until consensus was reached.

From the identified categories, we recognized specific themes (subcategories) and the key ideas that would be the subject of analysis. For example, within the category of leadership, we needed to analyze the concept, style, and competencies (subcategories), and, more specifically, transformational leadership (see Table 2).

**Table 2.** Inductive content analysis: categories, subcategories, and key ideas.

| Categories | Preventive Culture | Leadership | Training and Learning |
|---|---|---|---|
| | Concept | Concept | Systemic nature of knowledge and preventive learning. |
| Subcategories | Advantages | Leadership style | Organizational learning in the framework of psychosocial leadership |
| | Challenges and obstacles | Leader competencies | Informal or on-the-job learning methodologies |
| | Responsibilities | | Interdisciplinarity of university training in OSH |
| Key Ideas | Safe culture and prevention: joint commitment to protect work | Transformational leadership: guiding safety and prevention progress | On-the-job training and learning: building an effective safety culture and training professionals prepared for new work challenges |

To verify the quality and validity of the study, we proceeded rigorously and comprehensively. First, we triangulated the sources of data to obtain a more complete and detailed understanding of the participants' perceptions and experiences. Second, to ensure the validity and relevance of the questions posed to the different focus groups, we based the development of the questionnaire scripts on the review of regulatory OSH laws and the analysis of the influential literature. Third, we checked the transcription of the interventions with the participants themselves to ensure the accuracy and reliability of the data. Finally, we selected expert individuals with extensive professional experience to achieve a specialized perspective and enrich the analysis of the results. Taken together, these

features contribute to greater validity and reliability of the study, and to obtaining solid and significant results regarding the role of leadership in the development and promotion of a preventive culture.

## 3. Results

We grouped the results into the main issues or topics that were identified in the transcription of the focus groups as key factors that can facilitate (or hinder) establishing flexible processes and procedures that effectively and simultaneously address the premises of health and productivity in organizations.

### 3.1. Preventive Culture

According to the transcription of the participants' statements, they understand preventive culture as a set of values, attitudes, norms, and behaviors that promote and encourage OHS in the company (ID6A-1: "Promoting a preventive culture in the company is key to achieving effective OHS management").

In addition, they consider that preventive culture implies that safety and health at work are considered a priority and are integrated into all processes and activities of the organization (ID4A-2: "Preventive culture must be part of the business strategy"). It also implies the conscious participation of all workers in the identification, evaluation, and control of occupational risks (ID7A-3: "Promoting preventive culture is a task that concerns everyone").

#### 3.1.1. Advantages of Preventive Culture

According to the participants, preventive culture has several advantages, including

- Cost reduction: Preventive culture can help reduce the costs associated with occupational accidents and illnesses (ID2A-4: "Preventive culture avoids costs to companies and society in general").
- Improved productivity: By reducing risks and improving working conditions, worker productivity can be increased (ID3A-5: "Preventive culture not only reduces accidents but also increases productivity").
- Improved company image: A company that promotes preventive culture can improve its image and reputation, which can attract customers and employees (ID5A-6: "Preventive culture can contribute to improving the company's image among workers and society").
- Improved work climate: A company that promotes preventive culture fosters a safer, healthier, and more enjoyable work environment for employees (ID1A-7: "When we take care of our employees, they take care of us . . . ").
- Reduced absenteeism: By reducing risks and improving working conditions, the number of work absences can be decreased (ID8A-8: "A safe and healthy work environment helps us reduce absenteeism and retain our talent").

#### 3.1.2. Challenges and Obstacles in Implementing and Promoting Preventive Culture

The participants highlight the challenges facing the implementation of preventive culture in companies, such as the lack of commitment of leaders and the lack of adequate training offerings. They note that preventive culture may require initial effort and investment (ID3A-9: "Preventive culture is a process that requires time, effort, and dedication"), but in the long term, it can have significant benefits in terms of health and safety at work, as well as in the productivity and competitiveness of the company (ID4A-10: "Preventive culture is an indispensable requirement for business competitiveness").

Likewise, they indicate that several obstacles must be overcome for preventive culture to become a reality, including:

- Lack of leadership: Leaders and managers must be committed to prevention and safety in the workplace, but in many cases, they are not. This may be due to a lack of knowledge about the benefits of prevention or a lack of resources to implement it.

- Lack of training and education: The lack of training and education in prevention and safety can be a significant obstacle to preventive culture. Workers and managers need to be informed about the risks and appropriate prevention measures.
- Lack of resources: The lack of financial resources and time can also be an obstacle to preventive culture. Implementing prevention measures can be costly and may require time and effort from the company.
- Resistance to change: In some companies, there may be resistance to change and the implementation of new prevention measures. Workers may feel that these measures are an additional burden and may be reluctant to change their way of working.

To overcome these obstacles, it is important for leaders and managers to commit to prevention and safety, to provide appropriate training and education to workers, and to allocate the necessary resources to implement adequate prevention measures. It is also important to address resistance to change and to promote a culture in which prevention and safety are a priority for all employees.

Some quotes that support what we just stated are as follows:

- ID7A-11: "If we want a safe and healthy company, we have to start from the top. Leaders and managers must commit to prevention and safety in the workplace and lead by example".
- ID2A-12: "Training and education are key to promoting a preventive culture. Workers and managers need to be informed about the risks and appropriate prevention measures to be able to take preventive action".
- ID6A-13: "Lack of resources can be an obstacle to prevention and safety, but we can also be creative and find effective and economical solutions. Sometimes, the best solutions are the simplest".
- ID5A-14: "Resistance to change is normal, but we can also turn it into an opportunity for positive change. By involving workers in decision-making and making them part of the solution, we can overcome resistance and promote a positive preventive culture".

Promoting a preventive culture is everyone's responsibility. According to the participants' opinion, all actors in the organization have a responsibility to promote a preventive culture (ID1A-1: "Preventive culture is not only the responsibility of the worker, but of the entire organization"), but leaders, managers, and directors play a crucial role in its promotion and consolidation (ID8A-2; "Leadership and the involvement of management are essential for preventive culture to take root in the organization"). They also highlight the importance of prevention services and ORP technicians in the promotion and dissemination of preventive culture.

*3.2. Leadership*

3.2.1. Participants' Conceptions of Leadership

It was identified that the participants have a conception of leadership rather than the leader. They propose different ways of conceiving leadership, among them:

- Leadership is a process: The participants agreed that leadership is a process, not a quality or trait of a person. It was emphasized that leadership involves a set of actions and behaviors that are carried out in a specific context and can be learned and developed.
- Leadership is a relationship: OHS experts suggested that leadership involves a relationship between leaders and followers. The importance of communication, trust, and collaboration in leadership was emphasized, and it was emphasized that leadership should be understood as a dynamic relationship between people.
- Leadership involves a shared vision: Participants noted that leadership involves a shared vision between leaders and followers. It was suggested that leadership should involve all members of the organization in defining the vision, mission, and values of the company, and that it should be consistent with the organization's goals and objectives.



- Leadership is necessary for promoting a preventive culture: OHS experts agreed that leadership is essential for promoting a preventive culture in the organization. It was suggested that leaders should be role models in ORP and should foster a culture of prevention and safety in the organization.
- The leader is important but not solely responsible: Although participants emphasized the importance of leadership in promoting a preventive culture, it was suggested that the leader is not solely responsible for OHS in the organization. The importance of the participation of all members of the organization in ORP was emphasized, and it was suggested that responsibility should be shared between leaders and followers.

In summary, our interviewees indicate that leadership in the context of OHS is a dynamic and relational process that involves a shared vision and is essential for promoting a preventive culture in the organization. Although the importance of the leader was emphasized, it was emphasized that responsibility should be shared between leaders and followers.

Among the most clarifying quotes of what we just stated, we selected the following:

- ID12B-9: "Safety and occupational health are a team effort. Leaders must lead by example, but all employees must feel part of the solution and be responsible for their own safety and that of their colleagues".
- ID16B-5: "Prevention and safety are fundamental to the culture of any organization. Leadership must promote a shared vision that involves all employees, from executives to front-line workers".
- ID14B-18: "Leadership in Occupational Safety and Health is a dynamic process that requires effective and constant communication. Leaders must be empathetic and understand the needs and concerns of their followers to foster an effective preventive culture".
- ID11B-24: "Leaders in Occupational Safety and Health must be aware that responsibility is shared. It is not just about implementing prevention measures but ensuring that all employees feel involved and responsible for their own safety and that of their colleagues".

We want to highlight the following experience because it illustrates well the point of view on leadership of the participating managers, which can be very useful for improving the understanding of leadership in the context of OHS and promoting a culture of prevention in organizations: "I haven't always been a boss. In my previous job, the team leader never seemed to care about our safety at work. He always told us to work faster, even if that meant taking shortcuts that increased risks. However, after a series of workplace accidents, the team leader radically changed his attitude. He began to ask us about our personal protective equipment, to conduct safety inspections, and to offer us training on ORP. His transformative leadership had a remarkable effect on the team, as we began to feel more valued and safer at work. We learned that leadership in OHS is not just about giving orders, but about genuinely caring for the wellbeing of employees and taking concrete steps to protect them" (ID18B-1).

### 3.2.2. Style and Leadership Role

Participants expressed that transformational leadership is crucial to promote health and labor productivity in a changing environment. Leaders must have a clear vision, promote innovation and change, and create a work environment that promotes collaboration, learning, and development. Some textual quotes in this regard are as follows: "We can't keep doing things the way we always have, we need leaders who dare to innovate and propose new ideas" (ID9B-3), "A good leader is like a gardener, they have to plant the seed of change and then nurture it, so it grows and develops" (ID17B-12).

In relation to the latter, they mentioned that clear and effective communication is essential to promote health and labor productivity. Leaders must provide accurate and timely information about changes in the work environment and the preventive measures necessary to maintain workers' health and safety (ID10B-4: "Communication is the key to success in any company, without it, we are lost!").

Likewise, participants mentioned that fostering resilience and emotional wellbeing is crucial to maintaining health and labor productivity in a changing environment. Leaders must promote a work environment that supports workers' emotional health and fosters collaboration and mutual support.

We support these ideas with the following textual quotes: "Workers' emotional health is just as important as their physical health, and leaders must promote both!" (ID13B-7), "Resilience is fundamental in a changing environment, we need leaders who can motivate their teams and keep them united in the face of challenges!" (ID15B-2).

### 3.3. Systemic Nature of Knowledge and Preventive Learning

We infer from the discussions of the participants the systemic nature that they attribute to preventive knowledge and learning. According to them, a detailed analysis of knowledge and learning management in relation to OHS requires considering the system by which such management is carried out. This system is fundamental to knowledge management and should be part of the company's overall system.

The preventive management system is complex and consists of different interrelated and hierarchical systems at the global, European, and, especially, national levels. Some examples of these systems include employment, work and prevention, public health, industrial safety, education, and research, as well as sectoral and local bodies.

Preventive thinking and knowledge are generated, maintained, and interact at these levels in a complementary way, both formally and informally, and are related to the job. In addition, preventive knowledge is interdisciplinary and influenced by social, legal, political, and economic factors.

To enhance learning within this system and its subsystems, it is necessary to carry out positive actions in both the public and private spheres or in public–private collaborations. The public implications of formal learning include the promotion and coordination of activities by the authorities. On the other hand, private implications refer to informal learning or learning adapted to specific needs and are part of the general duty to protect workers on the part of each employer, who must establish adequate working and training conditions and collaborate in public activities.

In summary, preventive learning and knowledge management are fundamental aspects that require a systemic approach. Cooperation between the public and private sectors is crucial to promoting safety and health in the workplace (see Table 3).

**Table 3.** Illustrative quotes and experiences: systemic knowledge and preventive learning.

| Characteristics | Quotes and Experiences |
| --- | --- |
| Importance of knowledge management and preventive learning in the work environment | "The other day at the factory, we realized that the safety training we received really helped us prevent an accident. It's amazing how preventive knowledge can make a difference on the job" (ID4A-12). "I never realized how much education influences our safety at work until I participated in a training course and saw how we could apply what we learned in our daily work" (ID2A-6). "The other day, a colleague told me how his experience in another country helped him identify a risk in our workplace. It made me think about how preventive knowledge has no borders and how we can learn from different cultures and approaches" (ID5A-20). |
| Need for collaboration between the different systems and actors involved. | "At the last company meeting, we talked about how the public and private spheres can work together to improve workplace safety. It was interesting to see how all the systems are connected and how they influence each other" (ID3A-4). "I was impressed to see how public-private collaboration in our area improved working conditions overall. It really shows that, together, we can achieve great things" (ID7A-15). "Our boss always reminds us that safety is everyone's responsibility. So, we share our experiences and learn from each other, which helps us adapt to the specific needs of each position" (ID1A-9). |

### 3.4. Organizational Learning in the Framework of Psychosocial Leadership

The participants consider that learning in the workplace is crucial for the development of both preventive and professional skills. Our experts argue that working safely and

efficiently go hand in hand, which is why it is essential to integrate preventive training with professional qualifications. Adult learning in the workplace, especially for those with experience, should focus on the practical application of new knowledge in the workplace and within the context of their acquired skills.

Preventive training should focus on aspects such as training and cooperative and participatory learning, as adults learn best when they are involved in problem solving and the development of critical thinking. To achieve this, an adequate work environment and an organizational approach that fosters motivation and mental wellbeing are needed.

Learning in organizations requires cooperation, interaction, and leadership that drives knowledge creation. It is essential to promote learning environments and encourage interaction between the individual and their environment through professional conversations and interactions. Knowledge management must address the creation of organizational memory and facilitate the flow of information for appropriate decision making.

Effective leadership is essential for the development and sustainability of knowledge and must be based on principles of emotional intelligence and mental wellbeing. Preventive management must be an indispensable element in organizational management, integrating with aspects such as the management dashboard, learning climate, learning structure, and access to external knowledge.

In conclusion, the interplay between cognitive abilities, human emotions, and the psychosocial context significantly impacts workplace safety, wellbeing, and efficiency. By adopting a psychosocial leadership and organizational learning approach, companies can foster a supportive and engaging work environment that promotes continuous learning and development. This holistic approach, as illustrated by the grouped real-life experiences of participants in Table 4, demonstrates the profound influence of combining these strategies on various aspects of the work experience, such as communication, collaboration, and employee engagement.

**Table 4.** Illustrative quotes and experiences: organizational learning and psychosocial leadership.

| Characteristics | Quotes and Experiences |
|---|---|
| Communication and Collaboration | "Our manager always encourages us to share our ideas and experiences in team meetings" (ID6A-21). <br> "By collaborating across the different levels of our organization, we were able to overcome communication barriers" (ID8A-3). <br> "I remember when we implemented 'mentoring' in our company. The guidance from more experienced employees really made a difference in how we adapted and learned in our work" (ID4A-19). |
| Organizational Learning and Safety | "In one training, we were taught to understand the context in which we work and how that affects our health and safety" (ID2A-10). <br> "We once had a training where the leader taught us how to apply critical thinking in our daily work. That helped us identify and address risks more effectively and improve our safety" (ID5A-22). |
| Psychosocial Leadership and Engagement | "Since our company adopted a psychosocial leadership approach, the work environment has greatly improved. We all feel more valued and are more committed to our work and job security" (ID13B-8). <br> "I've noticed that by encouraging interaction and support among colleagues, our company has created a space where learning flows naturally and helps us deal with work challenges more effectively" (ID9B-17). |

### 3.5. The Methodology Aimed at Informal Training or Experiential Learning

The participants highlight the need to use appropriate formal and informal techniques for informal learning, to prioritize this type of learning due to the experiential return from job performance and peer exchange, and to integrate it with occupational experiential and preventive learning management.

They suggest the following practical recommendations for the study's objective:

Prioritize informal learning: It is recommended to give greater importance to learning acquired through job performance experience and peer exchange, using both formal and informal techniques. For this, systematic reviews, professional exchanges, experiential learning, case studies, and problem solving are suggested.

Integrate learning management with ORP: It is recommended to integrate occupational experiential learning management with ORP, so that both are directed together towards achieving job and organizational objectives within a common policy framework. This involves implementing a preventive culture in the company and measuring the degree of job satisfaction and commitment through reliable indicators.

Evaluate and train personnel ergonomically and psychosocially: It is recommended to carry out a thorough ergonomics and individualized psychosocial evaluation and training that considers healthy learning linked to a significant job as a priority for human performance and wellbeing. This involves developing problem solving based on teamwork ergonomics and the ability for continuous learning at an individual, group, collective, and organizational level.

Plan the complementarity of training interventions and experiential learning promotion: It is recommended to plan the complementarity of training interventions and experiential learning promotion with those related to the rest of the preventive action, involving the target workers by their participation in the detection of training needs that address their personal and professional characteristics (see Table 5).

**Table 5.** Illustrative quotes and experiences about informal learning.

| Characteristic | Quotes and Experiences |
| --- | --- |
| Experiential learning and knowledge sharing | "In our organization, we have established spaces to exchange ideas and knowledge with colleagues from other departments. This enriches our work experience and helps us gain different perspectives on our work" (ID10B-5). |
| Experiential learning, adaptation and continuous learning | "Our focus on informal learning promotes continuous learning, enables us to adapt quickly to changes in the work environment, and helps us develop skills to meet new challenges" (ID15B-19). |
| Focus on informal learning | "In our company, we realized that learning from experience and sharing knowledge with our colleagues is key. We now focus more on informal learning and see how it helps us in our day-to-day work" (ID14B-7). |
| Encouraging experiential learning and job satisfaction | "With the support of our company, we have participated in workshops and activities that foster experiential learning. These experiences have not only helped us grow professionally and strengthen our commitment to safety and prevention but also increased our job satisfaction" (ID11B-13). |
| Encouraging participation | "I had the opportunity to participate in a professional exchange program, and it was an eye-opening experience. I learned a lot from my colleagues and was able to apply that knowledge in my work" (ID12B-3). |
| Needs-oriented training | "Planning training interventions in our company has been very helpful. It allows us to address our individual and professional needs and makes us feel more involved in the learning process" (ID16B-11). |
| Integrating prevention into learning | "By integrating occupational risk prevention into our learning, we have achieved a safer and more collaborative work environment, where we all support each other to achieve our goals" (ID17B-6). |

### 3.6. Interdisciplinary Approach to Risk Prevention at Universities

Participants consider that companies can shape their professional background through personnel selection and training, and the management of organizational and psychosocial factors. To achieve this, the commitment and training of leaders and management is necessary, as well as the support of public administrations. However, the training offered from universities and specialized institutions aimed at managers, executives, and technicians is unsatisfactory.

The lack of commitment from managers and the scarce presence of training content dedicated to their awareness are recognized problems in several European countries. In this context, experts suggest that educational reform should not be postponed any longer.

The integration of risk prevention in universities requires an interdisciplinary approach that combines independent preventive subjects with cross-cutting contents. These should be included in undergraduate and graduate degrees, especially in technical, educational, health, and business management areas.

To achieve preventive integration, we should promote complementarity among degrees, organizational interaction, and multidisciplinary teams, while enhancing university-acquired skills for resilient, healthy, and socially responsible organizations. We should address training deficiencies with a three-level intervention model:

- Level 1 (degree): Comprehensive multidisciplinary generalists.
- Level 2 (master): Safety and hygiene specialists.
- Level 3: Adapt other graduates' knowledge for advisory, managerial, or technical roles.

Implementing these changes will allow greater effectiveness in risk prevention and better integration of OHS in the university and business environment (see Table 6).

**Table 6.** Illustrative quotes and experiences about interdisciplinary focus in university risk prevention training.

| Characteristics | Quotes and Experiences |
|---|---|
| Safety audits and risk prevention standards | "We once had a safety audit and realized that we were not following any risk prevention regulations. That made us rethink the way we work and start complying with regulations" (ID25C-10). |
| Management commitment to training | "You know, folks, companies have the opportunity to improve the training of their staff, but there is a lack of management commitment and government support" (ID27D-6). |
| Effects of inadequate work demands | "In my last company, the boss made us work overtime without pay, and there was no risk prevention technician. In the end, many of us ended up sick or injured" (ID18C-19). |
| Risk prevention technician's roles and impact as a change agent | "The risk prevention technician should act as a change agent and mediator, and it would be helpful to clarify and complement their roles to improve the effectiveness of their actions" (ID33D-14). |
| Impact of risk prevention training | "I remember a company where I worked, where the risk prevention technician was really committed. He organized workshops and trainings for all employees, and that helped to reduce the number of occupational accidents. This example shows how a well-prepared and committed technician can make a difference in the safety and wellbeing of workers" (ID22C-7). |
| Positive impact of the creation of a risk prevention committee | "I remember that, at my old job, a risk prevention committee was formed, and from that point on, things improved. There was more communication and awareness of occupational hazards" (ID29D-3). |
| Training offer for managers | "It seems to me that the training offer for managers and senior managers is not up to par, and that affects risk prevention in companies" (ID21C-3). |
| Perception of university degrees | "I have noticed that current university degrees are not preparing professionals to manage resilient and healthy organizations" (ID28D-8). |

## 4. Discussion

This study aims at understanding the key factors that can facilitate (or hinder) establishing flexible processes and procedures that effectively and simultaneously address the premises of health and productivity in organizations. In doing so, we conducted focus groups with OSH experts and stakeholders.

The main themes that emerged during the focus group discussions can be summarized as follows: it is emphasized that the culture of safety and prevention is key to effective

management of ORP and requires integration into all processes and activities of the organization, as well as the commitment of both top management and workers. It is highlighted that transformational leadership is necessary to bring about the change towards a culture of safety and prevention, and training and learning are key to creating an effective culture of safety and prevention in any organization. Additionally, the importance of an integrated and collaborative perspective in preventive learning is emphasized, and there is a need to approach it from an interdisciplinary perspective. Finally, the importance of integrating risk prevention in the university is highlighted to train professionals prepared to face the challenges of today's labor market.

The results of this study coincide with previous studies that also highlight the importance of a culture of safety and prevention in the organization [5,16,17,43–45], the need for leadership to carry out positive changes in the organization [14,22,32], and the importance of training and learning to improve safety awareness and prevent occupational risks [10,15,40]. In a similar vein, Da Silva and Amaral [54] examined the factors that contribute to success and the barriers that hinder the process of implementing and evaluating occupational health and safety management systems (OHSMSs). They revised 21 articles published during the period 2007–2018 and concluded that, due to the cyclic nature of the existing OHSMS, the promotion of continuous improvement, training in occupational OSH, and leadership in the management of OSH are crucial factors in achieving success. On the other hand, they found the lack of epidemiological data and not being able to integrate occupational health and safety practices into their human resources management and business systems to be the main barriers for an effective implementation of OHSMS.

The main novelty of this study lies in the integration of these different dimensions into a broader framework of flexible processes that consider both health and productivity in a changing environment. The approach to the integration of risk prevention in the university can also be considered a valuable contribution to this field of study. Additionally, unlike most studies that focus on leader strategies or characteristics, this study focuses on the leadership process, which is especially suitable for a changing context where management tasks become more complex, interdependent, and volatile [14].

The conception of leadership as a process has the advantage of emphasizing that leadership is a dynamic activity that involves both the leader and the followers, and that evolves and changes over time. This approach recognizes that leaders not only need certain characteristics and skills, but also need to be able to adapt to different situations and contexts to achieve goals and improve their team's performance. Additionally, the conception of leadership as a process allows leaders and followers to take an active role in building effective leadership through interaction and mutual feedback. In summary, the leadership approach as a process offers a more dynamic and adaptable perspective that may be more effective for leading in complex and changing environments.

On the other hand, it should be noted that theories of organizational learning [28], knowledge management [20], personal and organizational resilience, and organizational culture are consistent with the joint results presented [32,36,37,55]. All these theories emphasize the importance of promoting a strong preventive culture, adequate training, adaptation, and effective leadership to improve safety and health in the workplace and reduce occupational accidents. Therefore, all of them can be useful within the proposed integrative approach.

The study uses a focus group approach, which is a valid and widely used qualitative tool in research to obtain information on participants' perceptions, opinions, and experiences regarding a specific topic [48–53]. Additionally, the study incorporates several strategies to ensure data validity and reliability: (a) Triangulation of data sources. By collecting information from various participants with different roles and experiences (public managers and business executives, prevention technicians, and university researchers), the study can offer a more complete and detailed view of the results obtained; (b) Development of questionnaire scripts based on the review of legal standards and relevant literature: this ensures the validity and relevance of the questions raised in the focus groups and

helps ensure that the topics discussed are pertinent and well-founded; and (c) Verification of intervention transcriptions by the participants themselves: This allows for ensuring the accuracy and reliability of the data obtained, as participants can confirm or correct transcriptions of their own interventions.

However, it is important to note that the validity and reliability of data in a qualitative study, such as the focus group, can be affected by various factors such as sample representativeness and size, group dynamics, and researcher bias. Although measures have been taken to improve the study's quality, the generalization of results may be limited due to intentional sampling, the relatively small number of participants, and the gender bias observed. In that sense, we want to emphasize the importance of promoting the inclusion of more women in the field of OSH, as our study shows a lack of female participants with extensive experience in this area. It is necessary to work towards eliminating barriers that prevent women from accessing and remaining in this field, promoting their training and education on equal terms. In addition, we urge companies and organizations to actively promote the hiring of women in ORP technical teams and OSH promotion, to ensure greater diversity and enrichment of gender perspectives in decision making and the implementation of preventive measures.

In addition, although the study addresses several issues related to promoting health and labor productivity in a changing environment, which allows for obtaining a broader and more holistic view of challenges and solutions in this area, further research is needed to deepen the results and continue advancing in this research direction. Some future steps that could be taken in this direction may be (a) analyzing the role of technology in ORP, including the use of tools and information systems for identifying, evaluating, and mitigating occupational risks; (b) evaluating the effectiveness of different training and learning approaches in ORP, including formal training, informal training, and experiential learning; or (c) investigating how the integration of OSH in university and professional training can influence workers' perception and behavior in relation to OSH.

Despite the inherent limitations of the study design, our findings offer some insight that can be used to improve practice and decision making in the field of ORP. First, the importance of building and developing a preventive culture for effective management of ORP is emphasized. Business leaders and advisers can work on integrating the preventive culture into all processes and activities of the organization, and the commitment of both top management and workers. This implies that promoting safety and ORP must be a fundamental value of the organization and not just a legal obligation.

Second, the implementation of a transformational leadership process is suggested to carry out the change that represents the implementation of a culture of safety and prevention. This implies promoting positive changes and fostering personal and professional development of workers, establishing clear objectives, promoting innovation, motivating employees, and fostering training and learning.

Third, the importance of investing in employee training and continuous learning to improve the organization's ability to identify, evaluate, and mitigate occupational risks is highlighted. Business leaders and advisors may consider investing in employee training and learning to improve employee safety awareness, leading to safer behaviors in the workplace.

Fourth, the need to address preventive learning from an integrated and collaborative perspective, where both authorities and companies and workers actively collaborate, is emphasized. Business leaders and advisors may consider promoting collaboration among all stakeholders to improve the effectiveness of ORP.

Finally, the importance of integrating risk prevention into universities to train professionals prepared to face the challenges of the current job market is suggested. Business leaders and advisors may consider fostering an interdisciplinary approach that combines independent preventive subjects with cross-cutting content so that ORP is a priority in all university degrees, especially in technical, educational, health, and business management areas.



The results shed light on several key aspects, such as the systemic approach to learning and knowledge management, the importance of psychosocial leadership and organizational learning, the need to promote experiential and informal learning, and the relevance of an interdisciplinary approach in university education in relation to ORP.

These findings can inform business leaders and government managers about the most effective practices and approaches to address emerging challenges in the work environment and to improve employee health and productivity. In addition, recommendations derived from the results can serve as a basis for the design and implementation of policies and strategies at the organizational and educational levels.

## 5. Conclusions

The main conclusions that can be drawn from the study are as follows:

1. The safety and prevention culture is crucial for effective management of OSH and requires integration into all processes and activities of the organization, as well as the commitment of both senior management and workers.
2. Transformational leadership is necessary to bring about the change towards a safety and prevention culture in the organization.
3. Training and learning are key to creating an effective safety and prevention culture in any organization.
4. An integrated and collaborative perspective is essential for preventive learning, and learning needs to be addressed from an interdisciplinary perspective.
5. The integration of risk prevention in universities is crucial to prepare professionals to face the challenges of the current job market.

These conclusions can be useful in guiding decision making and implementing effective OHS practices in organizations, as well as for training professionals in the university setting. Furthermore, the study provides a solid foundation for future research on OHS and the promotion of safety and health in the business and university environments.

**Author Contributions:** Conceptualization, J.M.-T. and J.M.L.-R.; methodology, J.M.-T. and J.M.L.-R.; validation, J.M.L.-P., D.C.-S. and J.M.L.-R.; formal analysis, J.M.L.-R. and J.M.L.-P.; investigation, J.M.-T. and D.C.-S.; data curation, J.M.L.-R.; writing—original draft preparation, J.M.-T. and J.M.L.-R.; writing—review and editing, J.M.L.-P. and J.M.L.-R.; visualization, J.M.L.-R. and J.M.L.-P.; supervision, J.M.-T.; project administration, J.M.-T.; funding acquisition, J.M.-T. All authors have read and agreed to the published version of the manuscript.

**Funding:** This research received no external funding.

**Informed Consent Statement:** Informed consent was obtained from all subjects involved in the study.

**Data Availability Statement:** In accordance with the principle of proactive responsibility and adhering to the obligations set forth by Spanish legislation (Law 3/2018, of 5 December) on Personal Data Protection and the guarantee of digital rights, the videos of the group sessions are not available in any public repository. However, they can be requested from the first author, always under contract that ensure the guarantees established by the European Union regulation.

**Acknowledgments:** We would like to thank ACESSLA (Asociación Científica de Expertos en Seguridad y Salud Laboral de Andalucía) for the administrative and technical support they provided for conducting the focus group sessions.

**Conflicts of Interest:** The authors declare no conflict of interest.

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
