# Peer review of "Leadership and the Promotion of Health and Productivity in a Changing Environment: A Multiple Focus Groups Study"

_safety, 2023_

Round 1

Reviewer 1 Report

I very much enjoyed reading this paper. It is clear in structure, the methods are well described, and the results section is comprehensive and well written. I believe that the paper will add knowledge and serve as a guidance for companies in their work with occupational safety.

The paper could do with a quick overview of the language. Sometimes the sentences are a little hard to comprehend due to length.

Author Response

Thank you very much for your positive review. We do hope that our manuscript is welcomed by the journal’s audience.

Reviewer 2 Report

I do think that this article has serious flaws and that additional experiments are needed.

More in detail:

- the study design is not clear enough and, given its merely qualitative nature, the reproducibility and thus the scientific soundness of the study appears to be very weak;

- therefore, the results are not sufficiently valid and the conclusions are not supported by such results;

- moreover, I found the originality/novelty of the article as low, since this topic has already been addressed by many other articles/studies in the past.

Minor editing of English language is required.

Author Response

Thank you for your time to review this manuscript. We understand the reviewer’s reasoning, but we consider that our approach is a valid way to identify potential barriers and opportunities for improving the effectiveness of OSH practices in the workplace. While our approach is based on a qualitative study using the focus group technique, we would like to highlight that we have followed the common criteria of originality, usefulness, and reliability for evaluating research in psychology (see references 1-8 for studies using a similar technique).

Furthermore, we consider that our study design is clear enough (see materials and methods section), and follows a qualitative approach that meets the necessary criteria for both code saturation (identification of new issues or topics) and meaning saturation (fully understand the identified issues or topics) [9]. Therefore, the validity of our results and conclusions are supported by the data gathered (169 pages of information, see comments of participants in the manuscript that support the topics identified by the qualitative software).

In addition, to ensure the reliability of our results, although we primarily used the focus group technique as the method of data collection, we also conducted reviews of relevant legislation and analyzed pertinent literature in the field of OSH. This combination of data sources allowed us to obtain a broader and enriching understanding of professionals' perceptions and experiences on the topic. Moreover, within the focus groups, we included participants with different roles and experiences, such as government managers, business executives, technicians, and researchers in OSH. This inclusion provided diverse and insightful perspectives, contributing to the validity and reliability of our findings.

Anyway, following your comments, we have divided the introduction into two sections to make clearer the contribution of the study and we have shortened the methods section to make clearer the design and procedure of the study. In addition, the manuscript has been proofread.

In conclusion, we would like to express our sincere gratitude for your comments and suggestions. We value your contribution and appreciate your commitment to research quality. We hope that our responses address your concerns satisfactorily.

References:

  1. Bolliger L, Lukan J, Colman E, et al. Sources of Occupational Stress among Office Workers-A Focus Group Study. International Journal of Environmental Research and Public Health. 2022;19(3).
  2. Dashtipour P, Frost N, Traynor M. The idealization of 'compassion' in trainee nurses' talk: A psychosocial focus group study. Human Relations. 2021;74(12):2102-2125.
  3. Lamb D, Cogan N. Coping with work-related stressors and building resilience in mental health workers: Acomparative focus group study using interpretative phenomenological analysis. Journal of Occupational and Organizational Psychology. 2016;89(3):474-492.
  4. Romero D, Flandrick K, Kordosky J, Vossenas P. On-the-ground health and safety experiences of non-union casino hotel workers: A focus-group study stratified by four occupational groups. American Journal of Industrial Medicine. 2018;61(11):919-928.
  5. Hoepner AG, MacMillan DG. Research on 'Responsible Investment': An Influential Literature Analysis Comprising a Rating, Characterisation, Categorisation and Investigation. SSRN. 2009.
  6. Meadows L, Morse J. The Nature of Qualitative Evidence. Thousand Oaks, California: SAGE Publications, Inc.; 2001.
  7. van Hoof J, Bennetts H, Hansen A, Kazak JK, Soebarto V. The Living Environment and Thermal Behaviours of Older South Australians: A Multi-Focus Group Study. International Journal of Environmental Research and Public Health. 2019;16(6).
  8. Haven T, Pasman HR, Widdershoven G, Bouter L, Tijdink J. Researchers' Perceptions of a Responsible Research Climate: A Multi Focus Group Study. Science and Engineering Ethics. 2020;26(6):3017-3036.
  9. Hennink MM, Kaiser BN, Weber MB. What influences saturation? Estimating sample sizes in focus group research. Qualitative health research. 2019;29(10):1483-1496.

Reviewer 3 Report

Dear Authors,

Your manuscript offers interesting and important findings for the readers of Safety. I would like to encourage you to improve the validity of your qualitative analysis by (a) explaining how many pages of transcripted data you have, (b) explaining that the quotes shown in the article were transcribed to English, (c) adding a specific code after each quote indicating who the interviewee was. This way the reader could follow if several quotes were made by one individual or a small group of interviewees. It would also make visible the distribution of quotes per panel. (d) When you show your quotes (from line 275), please be explicit and write e.g. “Here's a quote from our interviewee (ID=1) from Panel A:” 

Moreover, the research question is shown in lines 144-147. Please streamline your Results and Discussion so that the reader can clearly see how your study contributes to the research question.

Other comments:

line 12: Abbreviation “OSH” is not used in the abstract. Delete.

lines 26-28: According to whom? Consider revising the very first lines of the article.

line 41: Abbreviation “PRL”. Explain.

line 211: What are “STEM Technicians”?

lines 239-240: Please explain how one should read Table 2. It’s unclear.

line 451: The titles of Table 3 are not correct?

Author Response

Thank you for your time to review this manuscript and your useful suggestions to improve its quality. We have followed all your suggestions and now the manuscript explains the number of transcripted pages (169) and how were transcribed into English (see data analysis section), and introduces quotes and identifies them with the participant’s code -again, thank you for this suggestion.

Furthermore, following your comments, we have divided the introduction into two sections to make clearer the contribution of the study (research question is shown in lines 89-92). Also, we have added at the beginning of the results the following introductory paragraph to link results with the research question: “We have grouped the results into the main issues or topics that were identified in the transcription of the focus groups as key factors that can facilitate (or hinder) establishing flexible processes and procedures that effectively and simultaneously address the premises of health and productivity in organizations.” Similarly, we have introduced the discussion with the following paragraph: “This study aims at understanding the key factors that can facilitate (or hinder) establishing flexible processes and procedures that effectively and simultaneously address the premises of health and productivity in organizations. In doing so, we conducted focus groups with OSH experts and stakeholders.”

Thank you again for your detailed comments and suggestions. We have included all of them in the manuscript. In addition, the manuscript has been proofread.

Round 2

Reviewer 1 Report

I appreciate the scope and content of this paper, and hope that it will reach relevant stakeholders and hope that the authors' will take this one step further in publishing their results in practical guidelines.

Author Response

(The authors gave the same response as above.)

Reviewer 2 Report

I do think that the manuscript has been sufficiently improved to warrant publication in Safety.

Author Response

(The authors gave the same response as above.)

Reviewer 3 Report

Dear authors,

The revised manuscript is very good and useful for several fields! Still something could be done to improve the quality of the manuscript: Please take a look at Table 5 (and Table 3) and check the location of a quotation mark (first row) and missing periods (rows 4-5; also Table 3).

Author Response

Thank you very much for your positive review. We have introduced the suggested changes in Tables 3 and 5. We do hope that our manuscript is welcomed by the journal’s audience.